# Analysing the Confinement Effect in Hollow Core Steel-Concrete Composite Columns under Axial Compression

**DOI:** 10.3390/ma14206046

**Published:** 2021-10-13

**Authors:** Antanas Šapalas, Andrej Mudrov

**Affiliations:** Department of Steel and Composite Structures, Faculty of Civil Engineering, Vilnius Gediminas Technical University (Vilnius Tech), LT-10223 Vilnius, Lithuania; andrej.mudrov@vilniustech.lt

**Keywords:** steel-concrete composite, confined concrete, physical tests, numerical analysis

## Abstract

Spun concrete technology allows manufacturing the reinforced concrete poles, piles, and columns with a circular hollow core. This concreting method ensures higher concrete density and strength than the traditional vibration technique and self-compacting concrete. This technology defines an attractive alternative for producing steel-concrete composite elements, allowing efficient utilisation of the materials due to the confinement effect. This study experimentally investigates the material behaviour of the composite columns subjected to axial compression. The experimental results support the above inference—the test outcomes demonstrate the 1.2–2.1 times increase of the compressive strength of the centrifugal concrete regarding the vibrated counterpart; the experimental resistance of the composite columns 1.25 times exceeds the theoretical load-bearing capacity. The proposed mechanical-geometrical parameter can help to quantify the composite efficiency. The parametric analysis employs the finite element model verified using the test results. It demonstrates a negligible bond model effect on the deformation prediction outcomes, indirectly indicating the steel shell confinement effect and confirming the literature results.

## 1. Introduction

Steel and reinforced concrete are typical materials for compressive structural members. However, such a decision comes with additional costs because of certain advantages and drawbacks of these materials. For example, steel structures are lightweight, but their thin-walled cross-sections are susceptible to local buckling. Besides, the high thermal conductivity of steel and insufficient fire resistance make such structural systems unacceptable in many cases without additional fireproofing. On the other hand, concrete structures are resistant to buckling and fire, but they are heavy and massive, taking up much usable space. However, the combination of these materials in the composite can solve the above problems [1,2]. The concrete-filled steel columns exemplify the structurally efficient solution [2,3,4,5] in which the steel shell confines concrete, while the concrete restrains local buckling of the steel cover [6]. By itself, the confinement results from the Poisson’s ratio difference between the steel and concrete when axial compression induces the radial stresses at the steel-concrete contact interface. The confinement effect increases the compressive strength and deformation capacity of the concrete core [3,7].

Combining different materials in the cross-section can further improve the composite properties. For example, fibre-reinforced polymers (FRP) [8] and stainless steel [9] immune to corrosion can form the external shell of the column. Alternative possibilities are to use fibre-reinforced concrete (FRC) and ultra-high-performance concrete (UHPC), which are inherently brittle in compression [10], even combining those with high-strength steel [11,12]. The reference [13] presented compression test results of FRP tubes filled with FRC. The test outcomes demonstrate that the confinement effect increases with increasing the column length. This study also investigated an even more complex example of the concrete-filled pultruded glass fibre reinforced polymer tubes externally strengthened with carbon fibre reinforced polymer (CFRP) sheets. The test results demonstrate that the CFRP strengthening altered the brittle failure of the column to ductile one. The above concept is applicable for various types of fibrous materials and structures (e.g., [14,15,16]).

Solid core composite columns were the object of most experimental programs reported in the literature, and only several studies investigated columns with the concrete hollow core sections [1,2,3,7,17]. Nevertheless, the experimental results demonstrated the 5% to 25% increase in load-bearing capacity compared to the solid core columns and the column weight reduction.

Spun or centrifugal casting is well-known concrete technology, producing circular hollow core cross-sections. When fresh concrete is compacted at high rotation speeds, centrifugal force separates water from the concrete, moving it to the rotation centre and decreasing the water to cement ratio, increasing the concrete strength. As a result, a circular cross-section form with a uniform thickness is shaped. The spun casting also increases the concrete density regarding vibration technologies, and self-compacting concretes [18,19].

Demolition and reconstruction of concrete structures produce large amounts of solid waste, which is expensive in landfills. Thus, the partial replacement of concrete aggregate with recycled concrete waste defines an attractive perspective [20]. However, this replacement raises the water demands, increasing the water-cement ratio. The spun casting technology can naturally reduce the residual water-cement ratio, remedying the mechanical concrete properties [19,21]. This technology also is an attractive alternative for producing steel-concrete composite elements, allowing efficient utilisation of the materials due to the confinement effect [3,7]. However, the adequate determination of this effect remains a fundamental problem in the structural design because of the absence of reliable material models of the confined composite.

This study experimentally investigates the material behaviour and structural performance of the hollow core steel-concrete composite columns. A finite element model is developed to illustrate the complex stress-strain state characteristic of the confined concrete. The model verified using the test results is also used for parametric analysis of the composite columns subjected to the axial compression.

## 2. Materials and Methods

This test program includes 18 hollow section 450 mm long centrifuged concrete columns, 15 spun concrete hollow section cylinders, and 15 vibrated concrete prisms (100 mm × 100 mm × 400 mm). The concrete specimens were used to investigate the efficiency of the centrifugal concrete technology expressed in terms of compressive strength. The spun concrete cylinders had the same geometry and were identical to the composite columns; the external steel shell was removed from the concrete core on the testing day.

Five different concrete mixtures were used to produce the test specimens. Table 1 describes the mix proportions for a cubic meter. This table also presents the initial and residual water-cement ratios, (W/C)_ini_ and (W/C)_res_; the spun casting caused the water reduction in the concrete. A Portland cement 42.5 N and coarse quartz sand were used. Table 2 and Table 3 describes the main characteristics of the sand.

Two types of steel tubes were used as shell elements of the composite columns. Cold bended from a cold-rolled low-carbon steel sheet (EN 10130: 2006) and welding with a straight longitudinal butt weld produced the 250 mm diameter and 2 mm wall thickness tubes. Additionally, hot-rolled tubes having a diameter of 159 mm and wall thickness of 4.8 mm (DIN 17145; 13 Mn6) were also tested. Five tension coupons longitudinally cut from the tubes determined the mechanical properties of the steel according to the standard (EN 10002-1) requirements. Table 4 summarises the tension tests results, where *d* = the tube external diameter; *t_s_* = the shell thickness; *E**_s_* = the elasticity modulus; *σ**_y_* and *ε**_y_* = the average steel yielding limit and corresponding strain; *σ**_u_* and *ε**_u_* = the average steel strength and corresponding strain.

A Kont-31 centrifuge with supporting rubber belts (Figure 1) was used to cast the composite hollow core columns and concrete cylinders. Figure 2 depicts the spun casting formwork. Three different speeds (414 rpm, 720 rpm, and 1020 rpm) for 4.2, 12, and 7 min were used, rotating the formwork filled with concrete.

Figure 3 shows the composite column cross-section. After finishing the spun pouring procedure and formwork disassembly, the hollow section columns were placed into the environmental chamber with a maintained temperature of 18–20 °C and 80–90% humidity until the testing day. Besides, three 100 mm × 100 mm × 400 mm vibrated concrete prisms and three concrete hollow section cylinders were also produced for each concrete mixture (Table 1). As mentioned in this section above, the concrete cylinders represented the core of the composite columns with removed steel shells. All the specimens were stored in the same environmental conditions. Table 5 presents the main geometrical parameters of the composite columns. It also includes the average strength of the prisms (*f_pr_*) and concrete cylinders (*f_cyl_*) and the ultimate axial load of the column (*N_u_*).

The ratio between the experimental load-bearing capacity and the theoretical resistance (*N_exp_/N_th_*) can describe the composite efficiency [21]; the sum of all components’ contributions defines the resistance *N_th_*. Reference [21] shows that the composite efficiency is mainly dependent on the reinforcement ratio *ρ* = *A_s_/A_b_* and a mechanical-geometrical parameter *ψ* $=\left[\rho \cdot \sigma_{y}\right]/f_{cyl}$. The composite effect must exceed 1.0; it does not reach 1.3 for the considered composite columns with hollow concrete core (Figure 4). It can also be observed that the ratio *N_exp_/N_th_* decreases with the increase of the parameters *ρ* and *ψ*.

The ratio between the external and internal diameters of the concrete core *β_c_ = d_c,ext_/d_c,int_* is another characteristic parameter of the cross-section shown in Figure 3. All above parameters (i.e., *ρ*, *ψ*, and *β_c_*) are included in Table 5.

From the centrifuging process standpoint, 2.0 is the typical maximum and most efficient value of the coefficient *β_c_*, meaning that concrete has filled a 25% of the core. The further increase of the ratio *β_c_* can cause uneven distribution of the concrete inside the tube. Thus, this value was set for the graphs shown in Figure 4.

## 3. Test Results of the Composite Columns

Monotonic compression tests were carried out using a 5000 kN universal hydraulic testing apparatus P-500 (Armavir Machine-Tool Factory, Armavir, Russia ). Spherical hinges reduced the applied load eccentricity; the transversal restraining rings prevented the local failure of the supports (Figure 5). Longitudinal and transversal deformations were measured using eight 20 mm strain gauges, as shown in Figure 5. Figure 6 shows the average strain diagrams for each group of columns described in Table 5; a dashed line indicates the theoretical resistance *N_th_*.

## 4. Numerical Simulations

The nonlinear simulations are carried out using the finite element (FE) software Atena V5.9 [22]. This software is well acknowledged in a broad field related to the structural application of cement-based and fibrous composites and high-strength steel [23,24,25,26,27]. The deformation problem was formulated based on the incremental constitutive law of materials. Isoparametric tetrahedral four-node finite elements with one integration point (Gauss integration) are used to model the concrete core and steel shell. The steel supports are modelled using isoparametric brick eight-node finite elements with eight integration points. An average finite element size is 5 mm. An ideal contact between the support plates and composite column simulates the restraining ring effect (Figure 5). Quarter of the column are modelled in the three-dimensional domain following the symmetry conditions, as shown in Figure 7. This figure also shows the FE mesh structure.

The constitutive model for concrete is based on the fracture mechanics for the tensile failure and the plasticity for the compressive failure. The CC3 DNonLinCementitious2 model offered by Atena is utilized. This fracture modelling approach is based on the classical orthotropic smeared crack formulation and crack-band model. It assumes the Rankine failure criterion and exponential softening [28]. The solution employs a return-mapping algorithm to integrate the constitutive equations that can handle cases when physical changes such as crack closure occur. The material model allows simulating concrete cracking, crushing under high confinement, and crack closure due to crushing in any material direction [22]. The necessary input parameters of the concrete, i.e., tensile strength, the modulus of elasticity, and the fracture energy, are assessed by *fib* Model Code 2010 [29] using respective compressive strength values from Table 5. Table 6 describes the assumed model parameters.

The bilinear, elastic-plastic von Mises material model with hardening (CC3 DBiLinearSteelVonMises) describes the behaviour of the steel shell. Table 6 defines the yield strength, elastic modulus, and hardening modulus assumed in the model. The experimental values (Table 4) determine the first two model parameters.

The load-deformation response of the column is simulated incrementally, applying a 0.04 mm vertical displacement to the top steel plate, as shown in Figure 7a. The deformation problem is solved in a 3D formulation using Paradiso solver by the full Newton-Raphson iteration procedure. Two simulations were carried out. The first analysis assumes a bond model between the concrete core and steel shell. The contact interface is modelled assuming the following parameters: the tensile *f_t_* and cohesion *C* strength of the bond are 0.3 MPa and 1.0; the normal *K_nn_* and tangential *K_tt_* stiffnesses are identically equal to 6.4 TN/m^3^; the friction coefficient *u* is 0.1. The second analysis adopts an ideal connection between the materials. The yielding of the composite section, i.e., significant deformation increase under small load increment, determines the failure of the FE models.

Figure 8 compares the simulated and experimental transversal and longitudinal strain diagrams of the steel shell of the selected specimen, where the average response shown in Figure 6e defines the reference. Figure 8 demonstrates a good agreement between the numerical predictions and experimental data (longitudinal and transversal strains) in the elastic deformation stage. The simplicity of the assumed failure model of the steel explains the differences in the ultimate deformation stage. Furthermore, the simulation results demonstrated that the bond model makes no sense on the numerical outcome. Thus, the further analysis considers only the models having the perfect bond model.

Figure 9 exemplifies the application of the developed numerical model for assessing the cracking behaviour and the non-elastic deformation of the selected column specimen (considered in Figure 8). The figure also demonstrates the von Mises stress distribution in the steel shell at different loading levels.

## 5. Parametric Analysis

Four FE models are created to investigate the effect of the hollow core thickness on the deformation behaviour of the composite columns. The simulations assume the same modelling principles as described in the previous section. This parametric study considers a standard CHS 219 × 3 tube having the 219 mm external diameter, 3 mm wall thickness, and 450 mm length. Such columns are considered short (*l*/*d* ≤ 5.0) [21,30]. Therefore, they do not exhibit flexural buckling and fail predominantly due to bearing or local instability.

A steel S355 was selected for the tube. The bilinear isotropic von Mises model (the elastic modulus *E =* 210 GPa; the Poisson’s coefficient *υ* = 0.3; the yielding strength *f_y_* = 355 MPa; the tangential hardening modulus *E_h_* = 1 GPa) describes the material plasticity. The constitutive CC3 DNonLinCementitious2 model was adapted for the C40/50 concrete. This model includes non-linear behaviour in compression with hardening and softening, biaxial strength failure criterion, reduction of compressive strength and shear stiffness after cracking. Compressive strength of 40 MPa was set; the remaining input parameters of the concrete, i.e., tensile strength, the modulus of elasticity, and the fracture energy, were assessed by *fib* Model Code 2010 [29]. Figure 10 shows the considered cross-sections, covering the full range of technologically possible concrete layer thickness.

Table 7 summarises the parametric analysis results. Figure 11 shows the simulated transversal and longitudinal strain diagrams of the steel shell and concrete core. It can be observed that the concrete deformations exceed the steel shell strains; that could be a consequence of cracking of the concrete already observed at early loading stages. The ultimate load increases with a decrease of the reinforcement ratio, i.e., with an increase of the concrete core thickness. Remarkable, the corresponding increase of the composite effect, expressed in terms of the *N_exp_/N_th_* ratio in Table 7 (where the FE simulations determine *N_exp_*), is observed.

## 6. Discussion

The composite effect (Figure 4) approaches its maximum when the ultimate stresses are reached in all cross-section constituents contemporarily. This study employs the model [21,30] to estimate the effectiveness of the considered specimens expressed in terms of *N_exp_/N_th_*. Figure 12a shows the corresponding diagram of the experimental columns. The lines define the theoretical models drawn for the specimen groups S1–S4 and S5 (Table 5). The following main observations from this graph can be made:Concerning the specimens with cold-bended shells, the efficiency of the S2 and S4 specimen groups agree with the theoretical model. The remaining composite columns (i.e., S1 and S3 groups) were not efficient.The specimens with hot-rolled shells demonstrate exceptional efficiency on average 1.45 times exceeding the theoretical estimation.

The composite specimen series S2, S4, and S5 exhibited a ductile failure. On the contrary, the local buckling of the shell caused the failure of the specimens of series S1 and S3. That explains the efficiency results of Figure 12a. Besides, the steel shell strength governs the mechanical performance of the steel-concrete composite columns (Table 4).

At a specific thickness, local buckling can appear before the steel strains reach the yield limit. Therefore, the design codes for steel and steel-concrete composite structures (e.g., [31]) introduce the slenderness limit of thin-walled cross-section components to prevent premature failure. The limitation is based on the concept of effective cross-section area—the mechanical behaviour is assumed as plastic, plastic with limited rotation capacity, elastic, or elastic with reduced effective cross-section, depending on the minimum slenderness of the cross-section components. For plastic behaviour, ensuring the best performance, the diameter to thickness ratio of the circular tube must satisfy the following condition:(1)$$d/t<50\times \left(235/f_{y}\right).$$

Figure 6 shows the above limitation necessity. The maximum longitudinal strains of the specimen series S1 and S3 reached the 1.32‰ value that does not get the yielding strain limit of 2.00‰ (Table 4) because of the local buckling of the steel shell. The other two series (i.e., S2 and S4), made of the same tube, demonstrate opposite results—the ultimate longitudinal strains reached 2.20‰ and 2.50‰, respectively. That makes fully utilised steel strength. Almost doubled the concrete core thickness of the series S2 and S4 regarding the S1 and S3 columns (Table 5) efficiently improved the structural performance.

These results align with the parametric analysis outcomes (Section 5), though the FE simulations underestimate the ultimate performance (Figure 8). Figure 12b shows the efficiency evaluation results of the FE models (Table 7). The line indicates the theoretical predictions by [21]. The FE predictions correctly define the general trend, but the efficiency error is 10%. That is a consequence of the simplified material model of the steel specified in the Atena software. The study [32] supports this inference.

Thus, developing a reliable constitutive model is of primary interest and should be the further development object. The study also demonstrates that the local buckling of the hollow core composite column is sensitive to the d/t ratio (Equation (1)) and the thickness of the concrete core. Therefore, elaboration of the failure prediction formulas, taking into account the concrete thickness, is necessary.

The concrete infill has a negligible effect on the local buckling capacity of the investigated circular columns, even for specimens with manufacturing imperfections exceeding the current tolerances bound defined by Eurocode 3 [33]. Notwithstanding the differences between the failure’s mechanisms of the hollow tubes and their filled counterparts, the failure loads’ variation was insignificant. The maximum buckling load difference does not exceed 6% (Table 7). The failure localisation place does not also affect the result. These observations support the expectation that the inward deformations’ restrain does not increase the local buckling stress in the steel shell. Besides, the perfect bond assumption does not affect the simulation results, indirectly indicating the steel shell confinement effect and confirming the literature results [34].

Further spun concrete applications aim at developing efficient and fire resilient building structures employing high-strength steel and UHPC. The study [35] describes an attractive application of the hollow core columns filled with water for fire protection. Alternative solutions could employ corrosion-resistant FRP materials composing lightweight composite structures [13,14]. In any case, centrifugal concrete technology can substantially reduce the energy consumptions for the production of composite members that aligns with the current trends in material science [36]. The external shell of composite members is permanent formwork. It is not necessary to use steam hardening in such a case. Concrete can reach necessary strength in natural environmental conditions. Table 5 demonstrates that the spun concrete strength is 1.3–2.1 times higher regarding the vibrated concrete. The technological water reduction during the formwork rotation (Figure 1 and Figure 2) makes the efficient application of flowing concretes with high W/C ratios possible.

## 7. Conclusions

This study focuses on the composite behaviour of the hollow core steel-concrete elements subjected to axial compression. The results of the experimental program, which included 18 composite columns, were used for developing a reliable finite element (FE) model suitable for solutions to engineering problems and the development of efficient composite elements. The following conclusions were made:Spun or centrifugal casting technology allows the development of efficient composite elements using high W/C ratio concrete. The spun concrete strength is 1.3–2.1 times higher regarding the same vibrated concrete. On the other hand, the composite effect can exceed 25%, i.e., 1.25 times the theoretical load-bearing capacity.The local buckling of the steel shell in the hollow core composite column is sensitive to the tube diameter to thickness ratio and the concrete core depth. Thus, elaboration of the failure prediction formulas of current design codes is necessary, considering the concrete thickness.The predictions by the experimentally verified FE model correctly define the above tendency. Furthermore, the simulation results demonstrated that the bond model makes no sense on the numerical outcome, indicating efficient adhesion between the composite column constituents. However, the efficiency prediction error (underestimation) was 10%. That is a consequence of the simplified material model of the steel assumed in the modelling. Thus, developing a reliable constitutive model is of primary interest and should be further investigated object.The mechanical-geometrical parameter *ψ* ($=\left[\rho \cdot \sigma_{y}\right]/f_{cyl}$
, where ρ, $\sigma_{y}$, and $f_{cyl}$ are the reinforcement ratio, the steel yielding stress, and compressive concrete cylinder strength) can help to develop the efficient composite columns.


## Figures and Tables

**Figure 1 materials-14-06046-f001:**
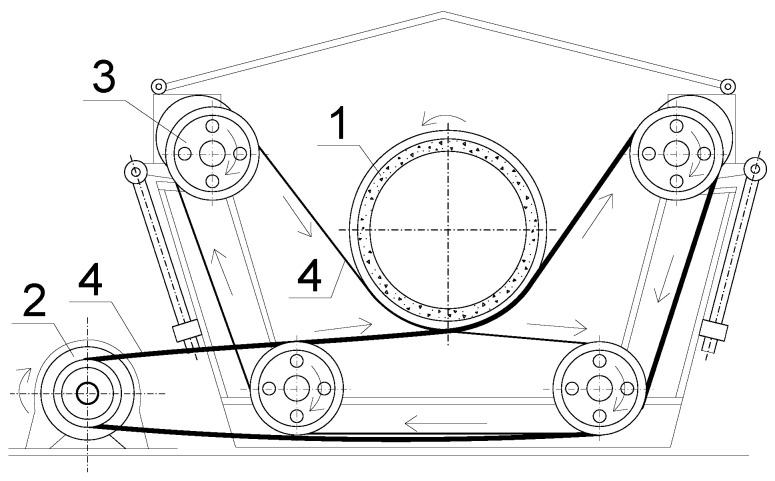
A Kont-31 centrifugal apparatus: (1) A tube filled with fresh concrete; (2) Engine; (3) Pulley; (4) Belt.

**Figure 2 materials-14-06046-f002:**
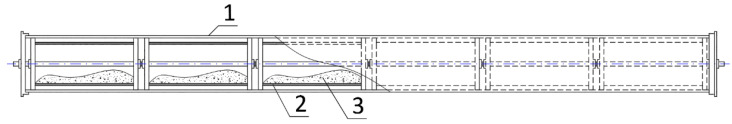
A formwork for the production of the spun specimens: 1 = The external formwork tube; 2 = The specimen tube; 3 = Concrete.

**Figure 3 materials-14-06046-f003:**
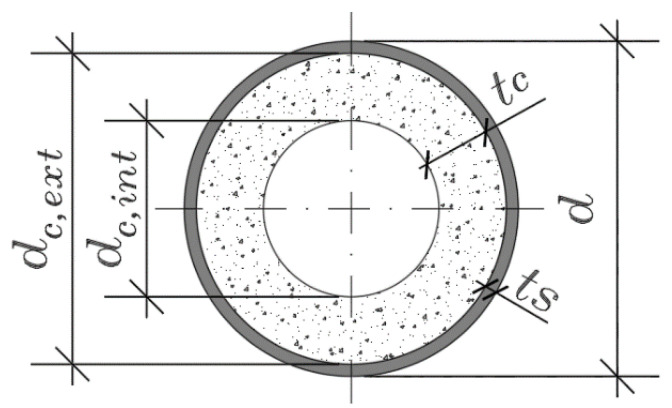
A cross-section of the hollow core composite column.

**Figure 4 materials-14-06046-f004:**
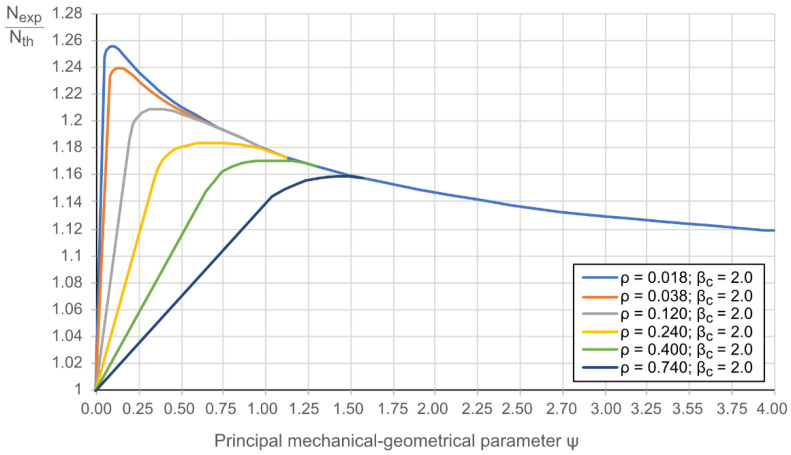
The composite effect in hollow-core columns, adapted from [21].

**Figure 5 materials-14-06046-f005:**
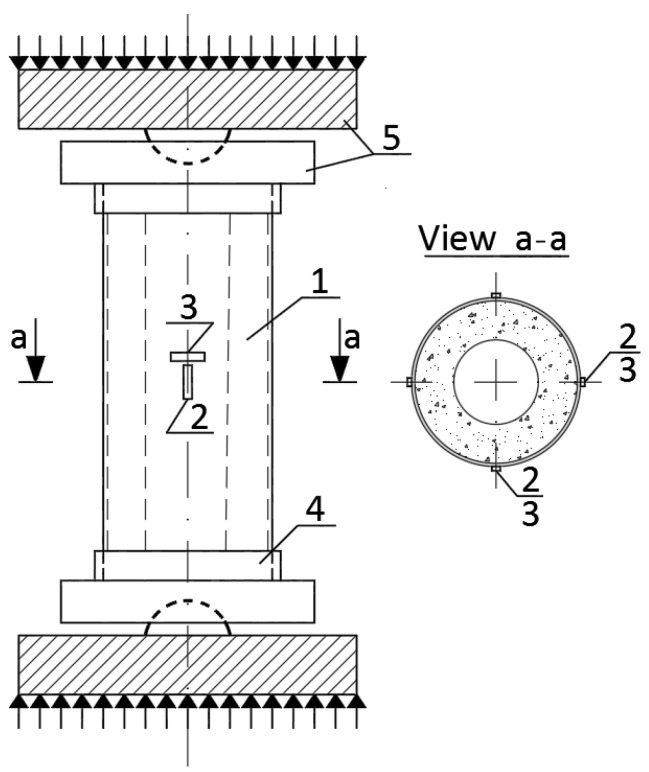
A schematic of the composite column test: 1 = Composite column; 2,3 = Longitudinal and transversal strain gauges; 4 = Transversal restraining rings; 5 = Supports of the testing machine.

**Figure 6 materials-14-06046-f006:**
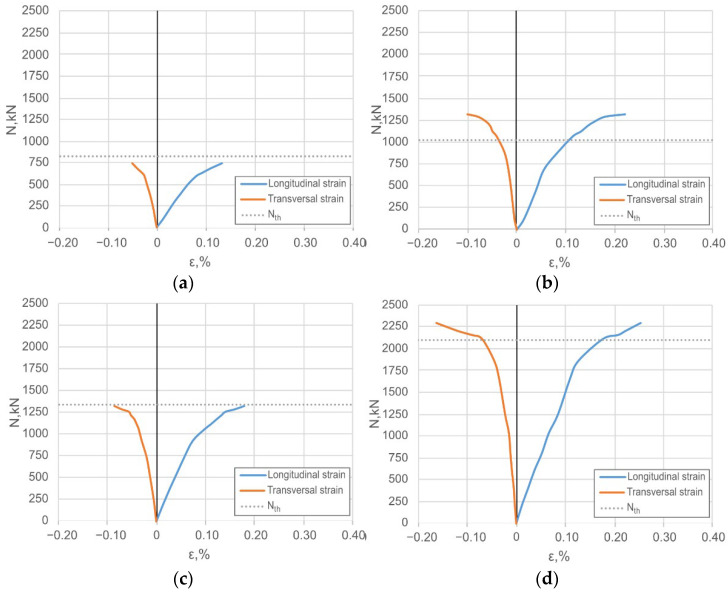
The average load-strain diagrams measured at the surface of the composite columns (Figure 5): (**a**–**e**) Corresponds to the results of the S1–S5 specimen series (Table 5).

**Figure 7 materials-14-06046-f007:**
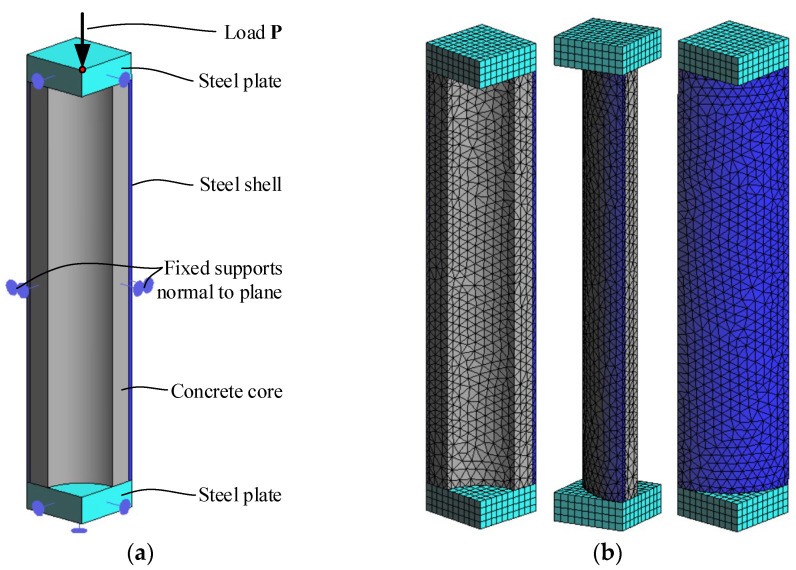
The finite element model: (**a**) Loading and boundary conditions; (**b**) FE mesh of the modelled column segment.

**Figure 8 materials-14-06046-f008:**
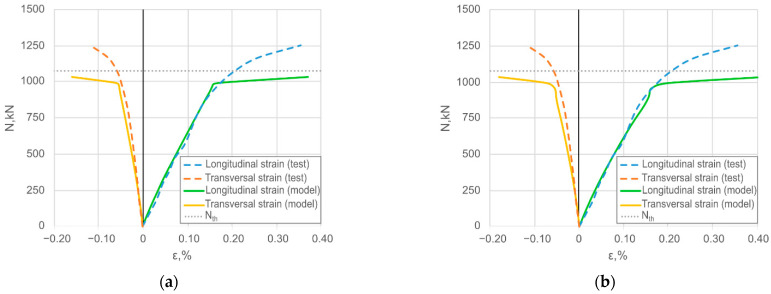
Verification example of the numerical model: (**a**) The simulations assuming a bond model between the concrete core and steel shell; (**b**) The perfect connection simulation result. **Note:** The simulation results are shown for the selected case when the average response shown in Figure 6e defines the reference.

**Figure 9 materials-14-06046-f009:**
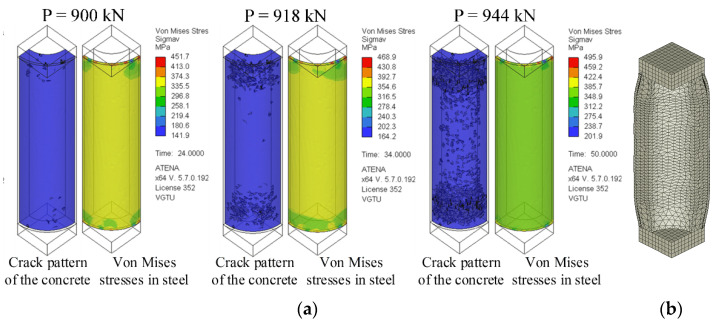
Numerical simulation results: (**a**) Crack pattern of the concrete core and von Mises stress distribution in the steel sheet at different loading levels of the selected specimen; (**b**) Deformed shape (vertical displacement = 2 mm) of the column model.

**Figure 10 materials-14-06046-f010:**
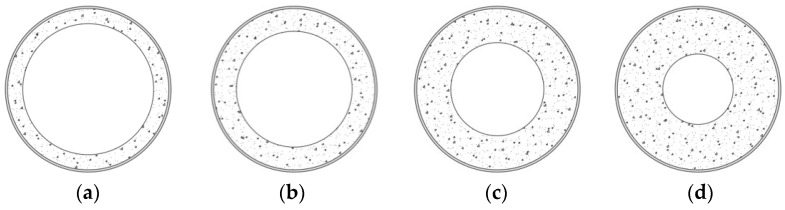
Parametric models of the hollow core composite columns: (**a**) A0 column with concrete core thickness *t_c_* = 20 mm; (**b**) A1 column with *t_c_* = 30 mm; (**c**) A2 with *t_c_* = 45 mm; (**d**) A3, *t_c_* = 60 mm.

**Figure 11 materials-14-06046-f011:**
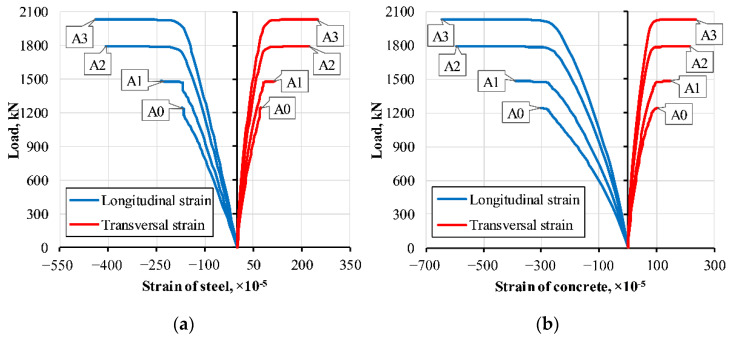
Numerically determined load-strain diagrams: (**a**) Steel shell; (**b**) Concrete core.

**Figure 12 materials-14-06046-f012:**
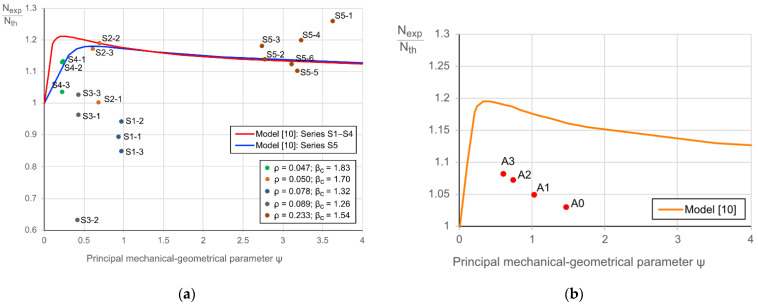
Assessing the composite effect in the steel-concrete columns: (**a**) Experimental specimens; (**b**) Numerical modelling results.

**Table 1 materials-14-06046-t001:** Concrete mixes proportions for m^3^.

Specimens Group	Weight, Kg/m^3^			(W/C)_ini_	(W/C)_res_
	Cement	Sand	Water		
S1	300	1705	282	0.94	0.55
S2	300	1705	270	0.9	0.53
S7	500	1464	270	0.54	0.34
S8	500	1464	260	0.52	0.34
S14	500	1700	330	0.66	0.43

**Table 2 materials-14-06046-t002:** Physical characteristics of the quartz sand.

Density, kg/m^3^	Bulk Density, kg/m^3^	Porosity, %	Impurity Level, %	Fineness Modulus
2500	1740	30.4	3.2	2.57

**Table 3 materials-14-06046-t003:** The sand granulometry.

Sieve Residuals, %	Sieve Mesh Size, mm					
	2.5	1.25	0.63	0.315	0.14	Passed 0.14
Partial	30.7	20.4	23.6	17.7	5.1	2.5
Full	30.7	51.1	74.7	92.4	97.5	–

**Table 4 materials-14-06046-t004:** Mechanical properties of the steel.

*d*, mm	*t_s_*, mm	*E**_s_*, GPa	*σ**_y_*, MPa	*ε**_y_*, %	*σ**_u_*, MPa	*ε**_u_*, %
250	2.0	210	288	0.17	365	17.6
159	4.8	215	350	0.20	460	19.0

**Table 5 materials-14-06046-t005:** Main properties of composite specimens.

Column	*d*, mm	*t_s_*, mm	*t_c_*, mm	*f_c,pr_*, MPa	*f_c,cyl_*, MPa	*N_u_*, kN	*ρ*	*ψ*	*β_c_*
S1-1	250	2	30.3	10.1	21.0	750	0.076	0.950	1.327
S1-2			28.9			775	0.079	0.989	1.307
S1-3			29.2			700	0.078	0.980	1.311
S2-1	250	2	48.6	14.0	19.6	1000	0.052	0.693	1.653
S2-2			47.5			1175	0.053	0.705	1.629
S2-3			55.5			1240	0.047	0.629	1.822
S3-1	250	2	25.0	17.0	52.5	1270	0.090	0.449	1.255
S3-2			25.9			850	0.087	0.436	1.267
S3-3			25.2			1360	0.089	0.446	1.258
S4-1	250	2	52.8	35.2	50.7	2300	0.049	0.252	1.752
S4-2			57.3			2400	0.046	0.238	1.872
S4-3			57.0			2200	0.046	0.239	1.864
S5-1	159	4.8	21.3	20.3	26.0	1300	0.271	3.637	1.399
S5-2			29.7			1250	0.208	2.791	1.660
S5-3			30.2			1300	0.206	2.756	1.679
S5-4			24.5			1270	0.242	3.243	1.488
S5-5			24.9			1171	0.239	3.201	1.500
S5-6			25.7			1200	0.233	3.121	1.524

**Table 6 materials-14-06046-t006:** Input parameters of the material models.

Material	Model	Strength, MPa		Modulus, GPa		Fracture Energy, N/m
		Compressive	Tensile	Elastic	Hardening	
Concrete	fracture-plastic	20	1.5	30	−	37.5
Steel	von Misesplasticity	350	350	215	1	−

**Table 7 materials-14-06046-t007:** Characteristics of the parametric models.

Series	Tube	*t_c_*, mm	*ρ*	*ψ*	*β_c_*	*N_exp_/N_th_*
A0	CHS 219 × 3	20	0.168	1.490	1.231	1.029
A1		30	0.118	1.048	1.392	1.049
A2		45	0.086	0.761	1.732	1.071
A3		60	0.071	0.626	2.290	1.081

## Data Availability

The raw data supporting the conclusions of this article will be made available by the authors, without undue reservation.

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
