# Peer review of "Analysing the Confinement Effect in Hollow Core Steel-Concrete Composite Columns under Axial Compression"

_materials, 2021, doi:10.3390/ma14206046_

Round 1

Reviewer 1 Report

The originality and the scientific value of the subject research are good.

The research area is Analysing the Confinement Effect in Hollow Core Steel-Concrete Composite Columns under Axial Compression.

The main part of the paper is focused on experimental tests and numerical modelling. 
A comprehensive view of the solution to the problem is very good and appropriate.
The solved area is current and very interesting for the readers of the journal of Materials.
The manuscript has the usual structure, including discussion and conclusion.
The introduction part must be fundamentally reworked and extended. Only 16 references are listed.

There is no clear motivation to solve the research problem and why the approach to numerical modeling was chosen. Extended the chapter and add at least 7 references.
Extensive research is underway in the area of experiments, nonlinear calculations, diagnostic of concrete structures and materials when it is necessary to rework and expand the information in the introduction section.
These are mainly the possibilities of material models of concrete, approaches to the choice of parameters, or taking into account the uncertainties in the calculation or stochastic character of materials.
Interesting research combines the field of use of reinforced concrete, diagnostics  and nonlinear calculation in the article:
Kozielova, M. et al. Numerical Analysis of Reinforced Concrete Slab with Subsoil. Civil and Environmental Engineering, 2020, 16 (1), 107-118.

The chosen methods in the manuscript for numerical modelling can be used and the research is solved logically.
However, very little information is provided on numerical modelling. It is also possible to find more advantageous approaches, but the chosen approach is possible.
However, more information must be provided.
The overall similarity of the calculation and the informative value must be substantially improved.
Provide detailed input parameters for the calculation, preferably in tables (material models, nonlinear solver parameters, ....)

Make adjustments:
1) Figure 1 - must be enlarged.
2) Figure 2 - must be enlarged. unable to read text.
3) Figure 3 - must be enlarged.
4) Figure 5 - must be enlarged.
5) Figure 7 - must be enlarged.
6) Figure 8 - must be enlarged.
Overall, the visual of the image and the informative value must be improved!

Overall, from the perspective of numerical methods, it is necessary to appreciate the knowledge and the chosen approach, but the detail and information value of the calculation must be improved.
It is also necessary to improve the discussion of the results and to set limits on experiments and calculations performed.
Overall, it is necessary to improve the informative value of the manuscript.
It is necessary to rework and expand also the introduction part (entry into the solved problem).
The manuscript must be revised before publication.

Reviewer 2 Report

The manuscript's theme "Analyzing the Confinement Effect in Hollow Core Steel-Concrete Composite Columns under Axial Compression" is interesting to Materials readers, however major changes are needed to the text for its approval.

a) The abstract must be completely reformulated. There are no quantitative results or conclusions reached in this study, which is experimental!!!

b) The introduction and consequently the state of the art is very limited and simple, authors should insert several other papers to support their discussions and increase their list of references, which is simple. I suggest some papers that must be included in the manuscript, but others must be inserted, such as: 10.1016/j.jmrt.2020.03.122; 10.32604/jrm.2021.015925; 10.1016/j.compstruct.2021.114672 (consider all indicated references and add others);

c) There are sequences of figures and tables that make the text tiring. You must introduce texts in order to discuss your results between tables and figures;

d) Discussions are limited and not based on studies of the literature on the subject, consider reformulating your discussions;

e) The highlight in terms of scientific innovation is limited to readers, please indicate this;

f) What future perspective of your research field? Indicate this in a paragraph of the conclusion.

Round 2

Reviewer 1 Report

The manuscript is reworked.

The manuscript is interesting for readers.

The manuscript contains all the important information.

Reviewer 2 Report

The authors did a great job and could be accepted for publication.